# Removal of Extended-Spectrum Beta-Lactamase-Producing *Escherichia coli*, ST98, in Water for Human Consumption by Black Ceramic Water Filters in Low-Income Ecuadorian Highlands

**DOI:** 10.3390/ijerph19084736

**Published:** 2022-04-14

**Authors:** Carlos Bastidas-Caldes, Juan Ochoa, Laura Guerrero-Latorre, Carlos Moyota-Tello, Wilson Tapia, Joaquín María Rey-Pérez, Maria Isabel Baroja

**Affiliations:** 1One Health Research Group, Biotecnología, Facultad de Ingeniería y Ciencias Aplicadas (FICA), Universidad de las Américas (UDLA), Quito 170125, Ecuador; 2Programa de Doctorado en Salud Pública y Animal, Universidad de Extremadura, 10003 Mérida, Spain; 3Ingeniería en Biotecnología, Facultad de Ingeniería y Ciencias Aplicadas (FICA), Universidad de las Américas (UDLA), Quito 170125, Ecuador; juanochoa007@gmail.com (J.O.); carlos.moyota.tello@udla.edu.ec (C.M.-T.); wilson.tapia@udla.edu.ec (W.T.); 4Catalan Institute for Water Research (ICRA), Water Quality Area, Emili Grahit 101, 17003 Girona, Spain; lguerrero@icra.cat; 5Unidad de Patología Infecciosa, Facultad de Veterinaria, Universidad de Extremadura, 10003 Mérida, Spain; reyjoaquin@gmail.com; 6Programa de Doctorado en Biología Molecular y Celular, Biomedicina y Biotecnología, Universidad de Extremadura, 10003 Mérida, Spain

**Keywords:** black ceramic water filters, water quality, resistant bacteria, extended-spectrum β-lactamase, ST10 clonal complex

## Abstract

Fecal contamination in natural water sources is a common problem in low-income countries. Several health risks are associated with unprotected water sources, such as gastrointestinal infections caused by parasites, viruses, and bacteria. Moreover, antibiotic-resistant bacteria in water sources have become an increasing problem worldwide. This study aimed to evaluate the bacterial pathogens present in water within a rural context in Ecuador, along with the efficiency of black ceramic water filters (BCWFs) as a sustainable household water treatment. We monitored five natural water sources that were used for human consumption in the highlands of Ecuador and analyzed the total coliforms and *E. coli* before and after BCWF installation. The results indicated a variable bacterial contamination (29–300 colony-forming units/100mL) in all unfiltered samples, and they were considered as high risk for human consumption, but after filtration, no bacteria were present. Moreover, extended-spectrum beta-lactamase-producing *E. coli* with *bla*_TEM_, *bla*_CTX-M9_, and *bla*_CTX-M1_ genes, and two *E. coli* classified in the clonal complex ST10 (ST98) were detected in two of the locations sampled; these strains can severely impact public health. The clonal complex ST10, found in the *E. coli* isolates, possesses the potential to spread bacteria-resistant genes to humans and animals. The results of the use of BCWFs, however, argue for the filters’ potential impact within those contexts, as the BCWFs completely removed even antibiotic-resistant contaminants from the water.

## 1. Introduction

Water pollution is a problem greatly impacting human health. Only an estimated 71% of the world’s population has access to safe drinking water [1,2]. Among the pollutants are bacteria, protozoa, viruses, and noxious chemicals. A study carried out in 2019 by the United Nations’ International Emergency Children’s Fund and the World Health Organization to assess water quality found that approximately 844 million people lacked a readily available source of safe drinking water. Of this number of people, 263 million spent more than 30 min per day finding water, and 159 million were using natural water sources for consumption. As a consequence of this lack of safe drinking water in rural areas, along with poor sanitation, the appearance of gastrointestinal diseases, such as infectious diarrhea, cholera, dysentery, hepatitis A, and typhoid, have become a serious human health problem [2].

Indeed, contaminated water has been estimated to cause approximately 502,000 deaths from diarrhea per year [3], while 10% of the deaths of children under five years old are related to diarrhea, with a total of 800,000 deaths per year [4]; these cases are clearly more frequent in developing countries. In Ecuador, diarrhea is the third leading cause of morbidity, along with 30,269 hospital discharges per year [5,6]. This frequency mainly occurs because of the very limited access to safe drinking water in rural areas. In fact, only 57.5% of the inhabitants have access to sanitation systems [7]. In addition, as a consequence of this contamination of water, multidrug-resistant bacteria, now globally widespread, represent a continuous risk of increased morbidity and mortality in the rural areas of developing countries [8].

The highlands of Ecuador, with rural villages and hamlets, is a prime example of this problem. According to the most recent data published by the government, only 52.1% of Ecuador’s inhabitants have access to a potable water system, while 47.9% must obtain this resource from wells, rivers, or rainwater [7]. Contamination from sewage, garbage, livestock, agricultural chemicals, debris, and other inputs, however, makes these alternative water sources unfit for human consumption. In addition, the data from the government specify that 19% of the inhabitants of the country do not have a correct mechanism for fecal elimination, with 7% eliminating waste into the same rivers that populations will later use for their consumption. Consequently, according to data from 2019, contaminated food and water were the cause of ca. 24,000 clinical cases [5,9].

Furthermore, in rural areas of Ecuador, certain antibiotics are used in animals for prophylactic treatment and growth promotion in unknown quantities [10,11]. These uncontrolled practices increase the selection for antibiotic resistance in the resident bacteria. These antibiotics are dumped into the rivers and ponds from which the inhabitants consume water. The antibiotics also reach the intestinal saprophytic microbiota in situ so as to reduce the effectiveness of those same drugs when they are subsequently used to treat gastrointestinal infections against the bacteria thus exposed [12]. This uncontrolled practice in recent years has led to the initiation of several types of research in Germany, Algeria, China, and Brazil, aimed at detecting bacterial-resistance genes, such as mobile colistin-resistant (*mcr-1)* and ESBL-producing *E. coli* in water samples [13,14,15,16]. In Ecuador, ESBL-producing *E. coli* and *mcr-1* genes were detected in the feces of yard animals and in urban fauna, as well as in human fecal samples [17,18,19,20]. In the Andean region of Ecuador, however, research on natural water sources has been very limited.

Within this context, a definite and sustainable solution that has been suggested for providing safe drinking water in low-income settings is the use of household water treatment [2]. The advantage of water-purification technology is its ease of use and its accessibility to remote communities. Among the devices that have been studied, compared to the classic models implemented worldwide, black ceramic water filters (BCWFs), new household water treatment devices that have been recently developed to improve microbial removal, were the best option. BCWFs have proved to be a viable alternative for eliminating bacteria and reducing viruses in contaminated water. Such filters, in addition to a complementary treatment, such as chlorination, can be used in homes to reduce enteric viruses and fully eliminate protozoa and bacteria, thus markedly diminishing the incidence of gastrointestinal diseases generated by contaminated water [21,22].

Consequently, BCWFs could be readily used for water purification among populations with limited access to safe drinking water. The present study was thus aimed at evaluating the degree of bacterial contamination in the water; genotyping *E. coli*; surveying the antibiotic-resistance genes in ESBL-producing strains from the water sources; and, finally, verifying the feasibility of BCWFs as effective household water-treatment devices for low-income populations in the highlands of Ecuador.

## 2. Materials and Methods

### 2.1. Study Area and Water Sampling

The highlands of Ecuador are Andean environments that range from 2300 to over 4500 m above sea level. Livestock activities and rural settlements are common in this area. The study region was located in rural neighborhoods around Nono village (0003.9263 S, 07834.6171 W) in the Pichincha province, a mountainous area located 30 km from Quito, the capital of Ecuador, with 2.5 million inhabitants (Figure 1). Firstly, we performed a survey of 40 families in the villages to identify the sampling sites through consideration of the water sources that were used for drinking by the local inhabitants. The procedure, in brief, was as follows: From each location selected after survey, samples were collected from the water sources (ponds and rivers) of the Nono district. Next, 5 L were taken in water sampling polypropylene bottles and transported to laboratories at the University of the Americas for subsequent analysis. In addition, samples from each drinking water source were collected in order to assess the effectiveness of BCWF in filtering fecal bacteria. The BCWFs used were an *in-house* developed technology for household water treatment, published in a previous study and applied in Ecuadorian low-income communities [23] (technical specifications in Appendix A).

### 2.2. Physicochemical and Microbiologic Analysis of BCWF Efficacy

The samples from the five locations were first analyzed to evaluate the contamination present in the water sources for those communities. The characterization of physicochemical and microbiologic parameters of water quality was carried out through the use of standard methodologies [24] before and after BCWF filtration; 1 l from each natural water source was filtered through a BCWF for efficiency analysis. A Student t-test was performed in order to determine significant differences between before and after filtrating for physicochemical analysis. Microbial analysis consisted of quantification of the total coliforms and *E. coli* present. To that end, 100 mL of the water samples, before and after BCWF filtration, were filtered through nitrocellulose membranes of 0.45 µm pore size (Millipore, Amsterdam, The Netherlands) and incubated in a chromogenic agar (Chromocult Agar, Merck, Darmstadt, Germany) at 37 °C for 18 h. Typical colonies of *E. coli* and coliforms were counted and quantified [25] For the confirmation of *E. coli*, the colonies from the chromogenic agar were transferred to EMB agar and incubated at 37 °C for 24 h. Typical metallic-green colonies were selected for the indole and TSI biochemical tests. Bacterial cryopreservation was performed in brain heart infusion broth, with 10% (*v*/*v*) glycerol, and was frozen at −20 °C and −80 °C for further analysis.

### 2.3. Molecular Genetic Determination of Resistance Genes in E. coli

In order to detect the presence of resistance genes, the genomic DNA of *E. coli* was extracted with the PureLink^®^ Genomic DNA Kit (Invitrogen, Waltham, MA, USA), according to the manufacturer’s instructions. The resulting DNA samples were quantified in NanoDrop equipment (Thermo Scientific^TM^, Waltham, MA, USA, ND-2000) and stored at –80 °C. For genetic identification of *mcr* genes, a polymerase chain reaction (PCR) was performed for *mcr-1* with the forward, 5’-GCTACTGATCACCACGCTGT-3’, and reverse, 5’-AGCTGAACATACACGGCACA-3’, primers, yielding a product size of 698 bp. ESBL genotyping was performed through the use of a Qiagen Multiplex PCR-plus kit (Qiagen, Germantown, MD, USA) in a method that can simultaneously detect *bla*_TEM_, *bla*_SHV_, *bla*_CTX-M1_, *bla*_CTX-M2_, *bla*_CTX-M9_, and *bla*_CTX-M8/25_, as previously described [26].

Gene amplification was carried out in an Eppendorf thermocycler, Mastercycler^®^ PRO (Eppendorf, Hamburg, Germany). The conditions and cycles for thermocycling for the *mcr-1* gene were as follows: initial activation (1 cycle at 95 °C, 15 min), denaturation (94 °C, 30 s), annealing (62 °C, 90 s), and extension (72 °C, 90 s) for 30 cycles, followed by a final extension (1 cycle at 72 °C, 10 min). In addition, the conditions for ESBL screening were as follows: initial activation (1 cycle at 95 °C, 3 min), denaturation (95 °C, 30 s), annealing (60 °C, 90 s), and extension (72 °C, 90 s) for 30 cycles, followed by a final extension (1 cycle at 68° C, 10 min). The PCR products were analyzed on a 2% (*w*/*w*) agarose gel with SYBR Safe and 1X Tris-borate-EDTA buffer. Electrophoresis was programmed at 100 V for 30 min in a Labnet Enduro Gel XL horizontal chamber (Labnet International, Inc., Edison, NJ, USA). The agarose gel was visualized on a ChemiDoc^TM^ Imaging Systems photo-documentation system (BioRad, Hercules, CA, USA) by means of Image Lab^TM^ software version 5.2.1 (BioRad, Hercules, CA, USA). The size of each band observed in the gel was determined through comparison with a 100 bp DNA ladder. The positive control DNA for both *mcr-1* and ESBLs multiplex was kindly donated by the Osaka Institute of Public Health, Japan.

### 2.4. Multilocus Sequence Typing (MLST)

MLST was performed to determine the genetic diversity of *E. coli* isolates according to the MLST Pasteur database (http://pubmlst.org, accessed on 20 June 2021). The PCR conditions of seven housekeeping genes (*adk*, *fumC*, *gyrB*, *icd*, *mdh*, *purA*, and *recA*) were amplified and sequenced as described by Wirth et al. (2006). The PCR products were sequenced using the Sanger sequencing technique in an ABI 3500xL Genetic Analyzer (Applied Biosystems, Waltham, MA, USA) with a BigDye 3.1^®^ capillary electrophoresis matrix. The procedure, stated in brief, was as follows: the allelic profiling and sequence-type (ST) determination were also confirmed at the above PubMLST website. In addition, to further analyze the relationship between the different sequencing types, a phylogenetic analysis of the housekeeping genes was performed.

## 3. Results

### 3.1. Sampling Sites

Five natural water sources, three ponds and two rivers, were identified as being used for human consumption in the vicinity of the inhabited areas of Nono (Figure 1, Table 1). In the map of the Nono district (Pichincha province, Ecuador) in Figure 1, the numbers (1–5) indicate the location where the samples of water were taken, and the Universal Transverse Mercator coordinates of each sampling site are listed in the accompanying table.

### 3.2. Physicochemical Analysis of BCFW Efficacy

The physicochemical parameters indicated a difference in the percentage of the various components assayed before and after BCWF filtration of the water. All parameters tested showed significant levels of reduction after BCWF treatment, with the exception of nitrate, BOD, and alkalinity (Table 1).

### 3.3. Microbiological and Molecular Genetic Determination

Microbiological analyses before filtrating revealed the presence of *E. coli,* along with total and fecal coliforms at all the locations analyzed. Table 1 summarizes the counts of *E. coli* colonies in the water from each location before and after BCWF filtration. These findings point to significant bacterial contamination in the natural water sources that Nono’s population routinely uses for consumption. As for bacterial concentration, the locations with the highest bacterial presence of *E. coli* were the ones at the lowest altitudes, with >300 colony-forming units (CFU/100 mL) at sampling sites three and five, at 2439 and 2435 m above sea level, respectively (Appendix A). Furthermore, molecular genetic analyses revealed the presence of these three types of ESBL genes: *bla*_TEM_ and *bla*_CTX-M-1_ at site three, and *bla*_TEM_ and *bla*_CTX-M-9_ at site five (Appendix A 3.1). Both *E. coli* isolates were determined to be of ST 98 in the bacterial lineage referred to as clonal complex (CC) ST10 (Table 2). No isolate was positive for the *mcr-1* gene (Appendix A).

## 4. Discussion

To the best of our knowledge, the study reported here is the first to report the existence of ESBL genes in the water consumed in rural areas of the highlands in Ecuador. Indeed, we detected three different ESBL genes: *bla*_TEM_, *bla*_CTX-M9_, and *bla*_CTX-M1_. Several studies have indicated that the finding of ESBL genes in *E. coli* isolated from water samples is common. In a study on water sources in Germany, Hetty Blaak (2014) identified *bla*_CTX-M1_ and *bla*_CTX-M15_ genes [28]. In addition, in the Białka river in Poland, *bla*_CTX-M_, *bla*_TEM_, and *bla*_OXA_ genes were detected in a river near sewage treatment plants [29]. A study of the water in Lebanese estuaries identified a high prevalence of *bla*_CTX-M15_ genes in the *E. coli* present [30]. In Ecuador, a study carried out by Ortega (2020) determined that the CTX-M-1 group, and specifically the blaCTX-M15 gene, was the most prevalent in the Machángara River in Quito; this finding agrees with the prevalence of ESBL genes in Ecuador, with that locus being the most common ESBL-encoding gene found in humans worldwide [31].

It is noteworthy that the previous studies were performed in the lowlands, where the bacterial load is normally greater because of a drain on the associated bodies of water through greater human activity and a higher population density. However, the finding of ESBL genes in highland isolates with relatively few inhabitants is indeed striking, although it is not rare. One possible explanation is that the source of that water contamination is both animal and human [28,32]. In environmental samples, the genes of the CTX-M-1 group are the most prevalent. Furthermore, the feces of wild animals, such as birds, have been thought to contribute to ESBL-producing *E. coli* in surface water [33,34,35]. Schierack et al. (2020), in a study on wild birds in Germany and Mongolia, reported that *bla*_CTX-M-1_ and *bla*_CTX-M-9_ genes were detected in the two respective countries, along with a direct correlation with the prevalence of human infections in those countries [36]. In contrast, the genes of the TEM group are known to be more common in human environments; this association implies either a direct or indirect human contamination of the water sources in our study—and this is very likely because of the absence of an adequate program for water contamination management to protect and conserve that resource [29,35].

No positive isolates for the colistin-resistance gene, *mcr-1*, were found in the *E. coli* bacteria present in the water samples of this study, even though the inhabitants had been found to use this antibiotic as a fattening supplement in the cattle feed. Moreover, cattle graze in close proximity to the water sources that the population use for their consumption. Nevertheless, resistant pathogens may possibly be diluted in those natural water sources. Similar studies, however, have demonstrated the presence of *mcr* genes in water samples elsewhere. A study by Yang et al. (2017) detected *mcr-1* in *E. coli* and *Klebsiella pneumoniae* in a river in China [37]. Likewise, other investigations carried out by Hembach et al. (2017) found the resistance locus, *mcr*-*1,* in seven wastewater treatment plants in Germany [14]. The *mcr-1* gene can be present in aquatic environments because aquatic ecosystems, in general, are considered reservoirs of antibiotic-resistant bacteria. In Ecuador, a study conducted by Loayza-Villa et al. (2020) detected this gene in the feces of dogs and a chicken [17,20].

The finding that the MLST analysis of *E. coli* isolates indicated CC10 to be the predominant clonal complex in the water samples was of significance, as it agrees with other studies carried out in which ST10 CC was likewise predominant in extraintestinal environments, such as river water [31], wastewater [28], and food [38], as opposed to other clonal groups more commonly found in clinical samples, such as ST131 [39,40]. Furthermore, *E. coli* ST10 is widely found in intestinal samples from animals and humans as both a commensal and a pathogen. Although ST10 CC is commonly found to be susceptible to antimicrobials, the ability of this group to spread rapidly and survive in the environment is of concern [41]. ST10 CC is linked to multidrug resistance and ESBL and is recognized as an emerging lineage of foodborne extraintestinal pathogenic *E. coli* (ExPEC) [42].

In addition, within the overall diversity in the *E. coli* population, two lineages (ST10 and ST155) might function as reservoirs of the *mcr-1* gene; the larger of these sources was linked to ST10 [43]. Although *mcr-1* genes were not found in the present study, we consider the role of ST10 CC, as one of the main *E. coli* clonal complexes associated with animal enterotoxigenicity, to be highly relevant. In fact, Shepard et al. (2012) found that the majority of the porcine enterotoxigenic *E. coli* isolates belonged to three clonal complexes, 10, 23, and 165, and pointed out the relationship exhibited by ST10 CC to ESBL-resistance-associated elements [44].

This current ESBL presence could be involved in the increasing mortality rate, since patients with gastrointestinal infection do not present favorable responses to treatment with these antibiotics, which results in the clinical syndrome evolving into sepsis [45,46]. For the present study area, these findings point to a serious public health problem because the main cause of death for Nono’s inhabitants is related to gastrointestinal diseases, which are usually treated mainly with antibiotics; the frequency of this therapy can cause antibiotic resistance to increase progressively [5,23].

The use of BCWF in isolated populations that are without access to adequate sanitary systems is a suitable alternative recourse for water purification, and also represents an effective, environmentally friendly, and inexpensive device capable of preventing the health complications associated with multidrug-resistant bacterial infections. The hazard to human health was verified, since the initial physicochemical and biologic parameters of the water bodies tested were far outside the permissible limits [24]. Contamination with sewage water, garbage, and debris, as well as the leaching of chemicals involved in livestock production, have led to a marked deterioration of water quality and, therefore, the consumption of that water constitutes a serious health hazard [47,48]. BCWFs were effective in reducing the major contamination parameters in raw water, such as nitrites, biochemical oxygen demand, and chloride, by >70% (Table 1). Similar results have been obtained in several studies, where ceramic filters proved to be highly useful alternative tools for removing pollutants from water sources [23,48]. In the present work, the removal of bacteria by BCWF was 100% effective, as no bacterial colonies developed after filtration, including those carrying resistance genes. Finally, although the use of BCWF proved to be effective in the removal of bacterial pathogens, disinfection processes after filtering, such as chlorination, are recommended to ensure the quality of the water and the health of the people who consume it.

## 5. Limitations

The present study had several limitations. First, the survey was performed on a restricted number of natural water sources, though the sampling sites selected were representative sources of natural water that the inhabitants routinely used for their consumption. The geographical situation and the accessibility to the water bodies were limitations for the sampling of larger areas. Another limitation was that the origin of the fecal contamination was unknown, and to clarify that source would require more data from upstream and from higher ground. Due to the evidence of the use of colistin as a growth promoter in animals, the possibility of finding *mcr-1* genes in future studies remains pending. Moreover, a determination of whether the fecal contamination was of human, porcine, avian, or a wild origin—or a mixture of any of those possible sources—would also be necessary. To do so, molecular flocculation techniques would be required, along with isolation and identification of species-specific enteric viruses. Despite these limitations, we believe that the data presented in this study will assist in more completely understanding the status of the water for human consumption in the highlands of the Ecuadorian Andes and serve to make decisions for the implementation of the specific and affordable sanitary measures that we recommend for low-income populations.

## 6. Conclusions

The water sources for human consumption in the highlands of Ecuador exhibited major microbial pollution, including contamination with ESBL-producing *E. coli* that carried the *bla*_TEM_, *bla*_CTX-M9_, and *bla*_CTX-M1_ genes, and were detected in two of the five water sources. These findings represent a significant hazard to the health of the inhabitants of these highlands. Although the use of the antibiotic colistin as a growth promoter in animals was common in the study area, the samples screened did not manifest any presence of *mcr-1* genes. Nevertheless, two of the *E. coli* isolates belonged to the ST10 (ST98) clonal complex, which indicates the presence of these bacteria as a continuous potential risk in spreading multidrug-resistant enteric bacteria within these low-income populations. As a solution, the BWCFs made in Ecuador would be effective in the complete removal of these potentially dangerous bacteria, present in natural water sources, and undesirable chemical species. We therefore strongly recommend the distribution of BCWFs in all areas with limited access to drinking water in order to reduce the incidence of diseases associated with the use of contaminated water. It is necessary to deepen studies into the contamination of natural water sources that are used for human consumption by using molecular and phylogenetic techniques that would allow us to determine the origin of the contamination. This is important since natural water sources are the only ones that many inhabitants of low-income rural communities have access to.

## Figures and Tables

**Figure 1 ijerph-19-04736-f001:**
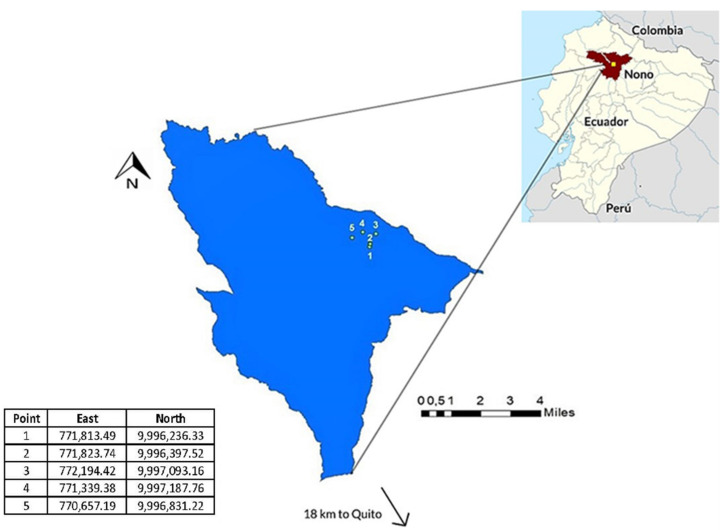
Map of Nono district, Pichincha province, Ecuador. The overall figure illustrates the geographical orientation of Ecuador within Central America. In the expanded map of the inset, the numbers (1–5) indicate the locations where the samples of water were taken, with the Universal Transverse Mercator coordinates of each point listed in the table below.

**Table 1 ijerph-19-04736-t001:** BCWF bacteria-removal assay with natural water testing, physicochemical removal in filtered water on the basis of drinking water guidelines [27].

Parameter	Before BCWF	After BCWF (*n* = 5)	SD (After BCWF)	Reduction Level (%) After BCWF	*p* Value
Turbidity (NTU)	3.38	1.72	1.83	49.11 *	**0.02138**
Nitrite (mg/L)	0.13	0.02	0.02	85.18 *	**0.02708**
Nitrate (mg/L)	1.77	1.54	0.60	13.39 ^NS^	0.07752
BOD	17.80	2.80	1.30	84.27 ^NS^	0.1226
COD	34.41	10.65	1.88	69.06 **	**0.00475**
Ammonia (mg/L)	0.11	0.04	0.02	63.70 **	**0.002501**
Phosphate (mg/L)	1.84	1.13	0.05	38.49 ***	**0.00018**
Chloride (mg/L)	58.46	17.53	8.54	70.01 ***	**0.00007**
Sulfate (mg/L)	14.74	9.84	5.91	33.26 **	**0.00678**
Oil (mg/L)	0.01	0.01	0.00	58.90 *	**0.03597**
Alkalinity (mg/L)	131.19	109.29	36.93	16.69 ^NS^	0.05738

Significance level: * 0.05; ** 0.01; *** 0.001; ^NS^ no significance. Bold *p*-values indicate significant differences obtained by the Student *t*-test of parameter-reduction indices upon the use of BCWF for water purification. BOD, biochemical oxygen demand; COD, chemical oxygen demand; SD, standard deviation; NTU, nephelometric turbidity unit.

**Table 2 ijerph-19-04736-t002:** Comparison of the microbiological characteristics of the water before and after the filtration with black ceramic water filters.

Sampling Points	Natural Source	Distance to the Town Center (Km)	Altitude (Masl) *	CFU/mL before Filtering	ESBL-Producing *E. coli*	Clonal Complex (Strain)	CFU/mL after Filtering
1	Pond	4.07	2594	29	-	ND **	0
2	River	4.2	2583	31	-	ND	0
3	Pond	4.98	2429	>300	*bla*_TEM_, *bla*_CTX-M-1_	ST10 (98)	0
4	Pond	4.72	2623	74	-	ND	0
5	River	4.18	2435	>300	*bla*_TEM_, *bla*_CTX-M-9_	ST10 (98)	0

* Meters above sea level; ** ND, not determined ST.

## Data Availability

Any data, photographs, coordinates, sequences, or analysis will be made available to readers upon reasonable request.

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
