# Peer review of "Removal of Extended-Spectrum Beta-Lactamase-Producing Escherichia coli, ST98, in Water for Human Consumption by Black Ceramic Water Filters in Low-Income Ecuadorian Highlands"

_ijerph, 2022, doi:10.3390/ijerph19084736_

Round 1

Reviewer 1 Report

Dear Authors,

The manuscript evaluated the efficiency of black ceramic filters to remove Escherichia coli ESBL present in natural water sources used for human consumption. The paper brings out an exciting subject and deserves more attention. The present study highlights a lack of knowledge of water quality consumption concerning resistant bacteria, especially in remote communities from low-income countries. The manuscript is clear, well-structured, and executed. However, some points need to be clarified.

Title. "Extended-Spectrum Beta-Lactamase–Producing Escherichia coli ST98 in Water of Human Consumption: Protection by Black Ceramic Water Filters In Low-Income Ecuadorian Highlands". I suggest being more concise.

Materials and Methods

Line 108 "From each location, 5 l were taken in plastic bottles for subsequent analysis". Only one campaign was made on each site? Or another biological replication was done? Please, clarify this point. If only one were made, this information must be on the text.

Line 113. "BCWF". Introduce a filter description.

Line 117. "standard methodologies".  The citation is missing.

Results

Line 178. "The parameters with significant reduction". Include the statistical analysis at the Materials and Methods topic.

Line 182. Term "effluent" at Table 1. Effluent is used to define the wastewater after treatment, and it is not very acceptable to use with water treatment. Replace this word for one more adequate to the context.

Discussion

Line 219. "In contrast, the finding of ESBL genes in highland isolates with relatively few inhabitants is indeed striking". It is common to detect ESBL genes in remote sites such as the Antarctic region. Please check 10.1128/AEM.07320-11

Line 232. "No positive isolates for the colistin-resistance gene, mcr-1, were found in the E. coli bacteria present in the Nono communities' water, even though the inhabitants have been found use this antibiotic as a fattening supplement in the cattle feed." Your research cannot confirm this statement with only five samples.

Limitations

Line 300. Include the data sampling campaign limitations.

Author Response

Dear reviewer

Reviewer 2 Report

Dear authors,

Many thanks for submitting the manuscript entitled ‘Extended-Spectrum Beta-Lactamase–Producing Escherichia coli  ST98 in Water of Human Consumption: Protection by Black Ceramic Water Filters In Low-Income Ecuadorian Highlands’ to the International Journal of Environment Research and Public Health.

I think that the manuscript has good quality, it is well written, well structured, and well referenced. Results were well discussed and some limitations of the study were included. It is an original work, especially as far as it's concerned the detection of antibiotic-resistant bacteria in the Andean region. One of my main concerns is about the very few samples analysed, the authors mention this in the ‘Limitations of the study’. Another point that needs to be improved is adding more technical details about the filters and also how the filtration was done? Was it in a household setting, was it in the Lab? The authors mentioned the limitation of filtration in reducing viral load, therefore ideally this research should have also assessed the virus removal efficacy (using bacteriophages as surrogates). This omission should be acknowledged in the manuscript and this should be included as a further research topic. Furthermore, the authors failed to mention the multiple barrier approach ( the importance of adding other treatments such as coagulation/flocculation prior to filtration and disinfection (chlorination or sodis/uv) after filtration). Please add this to your discussion especially the importance of chlorination after filtration and the importance of chlorine residual to kill or avoid re-growth of potentially damaged bacteria. Despite this, I think the manuscript deserves publication, especially of the novelty of detecting antibiotic resistance bacteria in the remote location of the Andes. Before publication, I think the authors should resolve the points above and those listed below.

line

Comment

26

What is significant here, it sounds vague? Need to compare to some sort of classification, see Lloyd and Bartram  (1991)

27

Which water volume?  Colony-forming units (CFU) per 100mL . 100 mL is how it is expressed WHO and EU guidelines!

28

E coli should be in italics, there are other places in the manuscript in which scientific names are not in italics, please correct.

30

This phrase is a repetition of the phrase 26

33-34

This phase is not very clear! Please re-write it why not change ‘clonal complexes’ by antibiotic-resistant bacteria.

34

THESE HHWTs? You have just mentioned one type of HHWT in this manuscript!!!

90-91

Should mention that further treatment (disinfection) should be employed after filtration to completely eliminate viruses

100-112

Ned to include much more details about the sampling procedure. How many samples, how many locations, how samples were transported, where water was filtered( lab or Household). Need to add some technical information of characteristics of the filters….

Where the microbiological analysis took place? Where the molecular analysis took place? 

Table 1

What do you mean ‘on the basis of drinking-water guidelines’? also, where is WHO 2011b in the reference list and add its numbered citation?

Table 1

Add a new column = (n) = the number of samples; Add turbidity unit; change effluent to ‘after filtration’ What is SD effluent? standard deviation?  Add meaning in the legend

193

300 CFU per 100 mL, 1mL ????

Table 2

First-line could be better formatted; Km appears twice,

275

What permissible limits? need references here

257-307

Need to include the importance of checking virus reduction after filtration, also need to add recommendations for further treatment (disinfection) after filtration

308

Need to mention Microbial source tracking techniques to try identifying the sources of faecal pollution

Author Response

Dear reviewer

Round 2

Reviewer 1 Report

Dear authors,

The manuscript's theme is relevant to the local and regional community, and the improvement highlighted the quality of your work.

 Regards